# Activation of XBP1 but not ATF6α rescues heart failure induced by persistent ER stress in medaka fish

Byungseok Jin[1] , Tokiro Ishikawa[1] , Makoto Kashima[2] , Rei Komura[2], Hiromi Hirata[2] , Tetsuya Okada[1] , Kazutoshi Mori[1]

The unfolded protein response is triggered in vertebrates by ubiquitously expressed IRE1α/β (although IRE1β is gut-specific in mice), PERK, and ATF6α/β, transmembrane-type sensor proteins in the ER, to cope with ER stress, the accumulation of unfolded and misfolded proteins in the ER. Here, we burdened medaka fish, a vertebrate model organism, with ER stress persistently from fertilization by knocking out the *AXER* gene encoding an ATP/ADP exchanger in the ER membrane, leading to decreased ATP concentration–mediated impairment of the activity of Hsp70- and Hsp90-type molecular chaperones in the ER lumen. ER stress and apoptosis were evoked from 4 and 6 dpf, respectively, leading to the death of all *AXER*-KO medaka by 12 dpf because of heart failure (medaka hatch at 7 dpf). Importantly, constitutive activation of IRE1α signaling—but not ATF6α signaling—rescued this heart failure and allowed *AXER*-KO medaka to survive 3 d longer, likely because of XBP1-mediated transcriptional induction of ER-associated degradation components. Thus, activation of a specific pathway of the unfolded protein response can cure defects in a particular organ.

## Introduction

Receptor-type transmembrane proteins at the plasma membrane and ligand-type secretory proteins are crucial for intercellular communication. These proteins reach their destination only after productive folding-mediated maturation in the ER. Therefore, failure in the quality control of these proteins in the ER and the resulting accumulation of unfolded or misfolded proteins in the ER, collectively termed ER stress, hampers various biological processes and may cause the development and progression of various diseases (Walter & Ron, 2011; Hetz et al, 2020).

The unfolded protein response (UPR), consisting of translational and transcriptional programs coupled with intracellular signaling from the ER to the nucleus, is activated to cope with ER stress in essentially all eukaryotes (Ron & Walter, 2007). The UPR is triggered via recognition of ER stress by three types of transmembrane proteins in the ER, namely, IRE1, PERK, and ATF6 in invertebrates, and IRE1α, IRE1β, PERK, ATF6α, and ATF6β in vertebrates (Mori, 2009). Activated IRE1, conserved from yeast to humans, initiates spliceosome-independent unconventional (frame switch-type) splicing of *XBP1* mRNA in metazoans to remove the 26-nucleotide intron, resulting in the production of the highly active transcription factor XBP1(S) (Yoshida et al, 2001; Calfon et al, 2002). Of note, we for the first time succeeded in the constitutive expression of XBP1(S) in medaka fish, a vertebrate model organism, by genome editing–mediated removal of the intron from the *XBP1* locus, and showed that the constitutive expression of XBP1(S) fully rescued the defects observed in *IRE1α/β*-double KO medaka; note that both IRE1α and IRE1β are ubiquitously and redundantly expressed in medaka (Ishikawa et al, 2017).

Activated PERK attenuates translation generally by phosphorylating the α subunit of eukaryotic translation initiation factor 2 (eIF2α) in metazoans (Harding et al, 1999). Paradoxically, attenuated translation leads to translational induction of the transcription factor ATF4, which induces genes involved in amino acid metabolism and resistance to oxidative stress, and the gene that encodes the proapoptotic transcription factor CHOP (Harding et al, 2000, 2003).

Both ATF6α and ATF6β are ubiquitously and redundantly expressed in medaka and mice (Yamamoto et al, 2007; Ishikawa et al, 2013). When activated in response to ER stress, ATF6α/β relocates from the ER to the Golgi apparatus where they are cleaved sequentially by Site-1 and Site-2 proteases in vertebrates, resulting in liberation of the cytosolic regions of ATF6α/β from the Golgi membrane as the highly active transcription factors ATF6α/β(N). These then translocate to the nucleus to enhance transcription (Haze et al, 1999; Ye et al, 2000; Nadanaka et al, 2004); ATF6α(N) is more active than ATF6β(N) as a transcription factor (Haze et al, 2001). Therefore, it is reasonable to consider that genome editing–mediated removal of the DNA sequences corresponding to the transmembrane and luminal regions of ATF6α from the *ATF6α* locus would lead to the constitutive expression of

[1]Department of Biophysics, Graduate School of Science, Kyoto University, Kyoto, Japan  [2]Department of Chemistry and Biological Science, College of Science and Engineering, Aoyama Gakuin University, Sagamihara, Japan

Correspondence: mori@upr.biophys.kyoto-u.ac.jp

ATF6α(N). Here, we show for the first time that this is indeed the case in medaka.

Activation of the PERK, ATF6, and IRE1 pathways leads to differential outcomes: PERK-mediated translational attenuation decreases the burden on the ER; ATF6-mediated rapid induction of ER-localized molecular chaperones and folding enzymes (hereafter ER chaperones) refolds unfolded and misfolded proteins accumulated in the ER; and ATF6- and IRE1-mediated induction of components of ER-associated degradation (ERAD) machinery degrades unfolded and misfolded proteins accumulated in the ER. These outcomes result in maintenance of the homeostasis of the ER (Yamamoto et al, 2007). If ER stress is further prolonged after activation of the UPR, the cell undergoes apoptosis (Mori, 2009).

Here, we asked what would happen if ER stress were evoked persistently and potently after fertilization in medaka. To this end, we focused on the *Meigo* gene, which encodes an evolutionarily conserved protein spanning the ER membrane multiple times. Its gene product was originally thought to function as a nucleotide sugar transporter and to play an important role in maintaining the homeostasis of the ER in yeast (Nakanishi et al, 2001), worms (Dejima et al, 2009), and flies (Sekine et al, 2013). Interestingly, however, its human orthologue SLC35B1 (Fig 1A) was recently shown to function as an ATP/ADP exchanger in the ER membrane (abbreviated AXER) (Klein et al, 2018; Yong et al, 2019; Kamemura et al, 2022). Thus, depletion of AXER reduces ATP levels in the ER lumen and consequently inhibits the activity of the Hsp70- and Hsp90-type major ER chaperones BiP (GRP78) and GRP94, respectively, evoking ER stress. Indeed, knockdown of AXER in HeLa cells decreased ATP concentration in the ER lumen to approximately one-third of that in control cells, but not to zero, suggesting that AXER is not the sole ATP/ADP exchanger in the ER (Klein et al, 2018).

We first investigated the phenotypes of *AXER*-KO medaka, all of which turned out to die by 12 days post-fertilization (dpf; medaka hatch at 7 dpf). We previously showed that a mutation in the ATP-binding site of BiP caused embryonic lethality at a very early embryonic stage (2–3 dpf) in medaka (Ishikawa et al, 2013). In contrast, all *AXER*-KO medaka live longer than 5–6 dpf. This indicates that ATP in the ER lumen is not completely depleted in *AXER*-KO medaka and that ATP is minimally present in the ER lumen, allowing BiP to function—albeit weakly—and other facets of ER biology to work under this condition.

We then asked whether the defects observed in *AXER*-KO medaka, namely, heart failure, could be rescued by the constitutive expression of XBP1(S) or ATF6α(N) from fertilization, namely, at the same time as the loss of AXER in medaka embryo (in other words, from the start of the decrease in the ATP level in the ER lumen), based on our expectation that active XBP1- or ATF6α-mediated induction of gene products would aid the maintenance of the ER protein homeostasis directly. In this connection, we previously showed that the constitutive expression of XBP1(S) from fertilization fully rescued the defects observed in *IRE1α/β*-double KO medaka (Ishikawa et al, 2017), as mentioned above, whereas in the present study, we show that the constitutive expression of ATF6α(N) from fertilization fully

rescues the embryonic lethality of *ATF6α/β*-double KO medaka, as expected.

# Results

## Induction of ER stress and apoptosis in *AXER*-KO medaka

To produce *AXER*-KO medaka, we introduced cleavage at exon 1 of the *AXER* gene using the transcription activator–like effector nuclease (TALEN) method by microinjecting TALEN-L and TALEN-R plasmids into one-cell-stage embryos, expecting the loss of the *Bsm*AI site (Fig 1B). The resulting G0 fish were incrossed to obtain F1 fish. A genomic PCR fragment obtained from one of 11 F1 fish showed resistance to *Bsm*AI digestion, and the mutation was transmitted to a germ line after crossing with WT fish. Resulting male and female F2 *AXER* +/− fish were incrossed to obtain F3 *AXER* −/− medaka. DNA sequencing revealed that two nucleotides were deleted at an expected position (Δ2, Fig 1B). This caused a frameshift at aa39, resulting in the production of a non-functional protein (Fig 1C). *AXER* −/− medaka exhibited a poor hatching rate (Fig 1D), and hatched *AXER* −/− medaka showed abnormal phenotypes at 1 day post-hatching (dph) compared with *AXER* +/+ medaka, namely, a smaller head with poor development of the skull and jaw, and a crooked and shorter tail (Fig 1E and F). All *AXER* −/− medaka died by 2 mo after birth (Fig 1G).

To monitor the level of ER stress in *AXER* −/− medaka, *AXER* +/− medaka were crossed with WT fish carrying the $P_{BiP}$-EGFP reporter, in which EGFP is under the control of the major ER chaperone BiP promoter ($P_{BiP}$), and whose fluorescence intensity well reflects the extent of ER stress (Ishikawa et al, 2011, 2013). Resulting male *AXER* +/− medaka carrying the $P_{BiP}$-EGFP reporter were crossed with female *AXER* +/− medaka and analyzed for fluorescence. Results showed that ER stress was markedly evoked in the entire body from 4 dpf in *AXER* −/− medaka compared with *AXER* +/+ medaka (Fig 2A). Quantitative RT–PCR revealed a significant increase in the level of BiP mRNA in *AXER* −/− medaka embryos from 2 dpf compared with *AXER* +/+ medaka embryos (Fig 2B), confirming the results of the fluorescence reporter assay. Thus, *AXER*-KO evokes persistent ER stress from an early embryonic stage.

To monitor the level of apoptosis in *AXER* −/− medaka, we employed VC3Ai, a genetically engineered Venus, which becomes fluorescent only after cleavage of the DEVD sequence inserted into Venus by caspase-like proteases (Zhang et al, 2013). VC3Ai was placed under the control of the β-actin promoter ($P_{actin}$) in an attB-targeting vector containing the tagCFP gene under the control of the zebrafish cardiac myosin light chain 2 promoter ($P_{zcmlc2}$) (Fig S1A(a)). After the expected phiC31 integrase–mediated recombination at the attP-landing site in chromosome 13 in WT fish with attB in the targeting vector (Fig S1A(a)), which was confirmed by genomic PCR (Fig S1A(b)), the ventricle exhibited tagCFP fluorescence from 3 dpf, in addition to mCherry fluorescence in the eye lens, which is under the control of the zebrafish heat shock protein 70 promoter ($P_{zhsp70}$) (Fig S1A(c)), as we described previously in Ishikawa et al (2018). Fluorescence from cleaved VC3Ai increased in response to treatment of WT fish carrying the VC3Ai reporter with staurosporine, a potent inhibitor of protein kinases, for 18 h (Fig S1B), or with

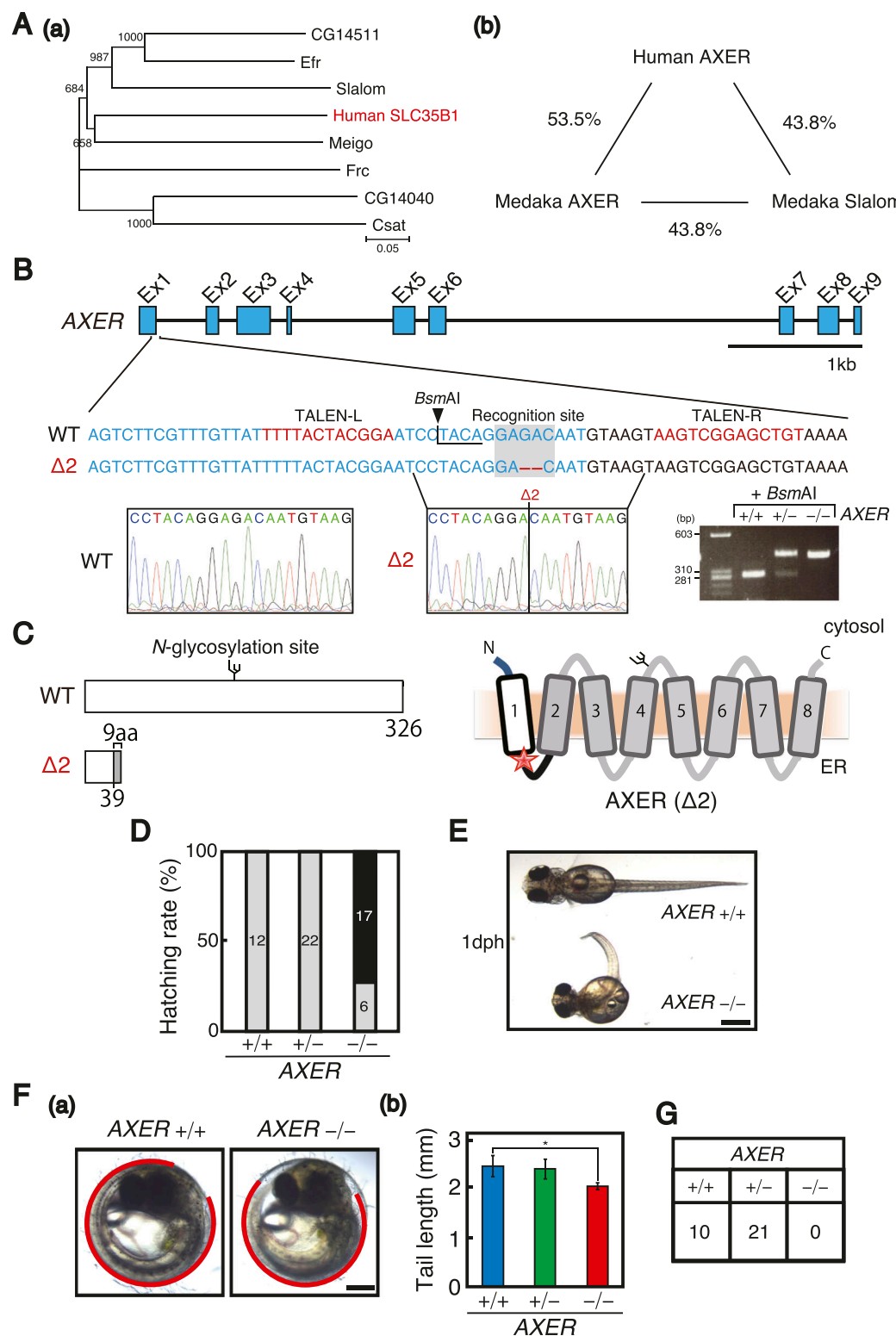

**Figure 1. Production and characterization of *AXER*-KO medaka.**
**(A)** (a) Phylogenetic relationship between medaka 7 putative nucleotide sugar transporters and human SLC35B1 (AXER) is shown. (b) Closer similarity between medaka Meigo (AXER) and human AXER than medaka Slalom and human AXER is shown. **(B)** Structure of the medaka *AXER* gene is schematically shown, and the sequences of a part of exon 1 of WT and Δ2 *AXER* genes (written in blue) are shown below together with the sequences of TALEN-L and TALEN-R (written in red), as well as the site of the *Bsm*AI digestion. PCR fragments amplified from *AXER* +/+, +/−, and −/− medaka were digested with *Bsm*AI and electrophoresed. **(C)** Structures of WT and Δ2 AXER proteins are schematically shown with the *N*-glycosylation site indicated. Δ2 causes a frameshift at aa39, which is followed by unrelated 9 aa and the stop codon. The resulting Δ2

tunicamycin, an ER stress inducer, for 30 h (Fig S1C). Note that the eye lens emitted green fluorescence because of the merging of mCherry and Venus fluorescence (Fig S1B and C); it is known that caspases are activated during lens cell differentiation (Zandy et al, 2005). *AXER* +/− medaka were crossed with WT fish carrying the VC3Ai reporter, and resulting male *AXER* +/− medaka carrying the VC3Ai reporter were crossed with female *AXER* +/− medaka and analyzed for fluorescence. Venus fluorescence in the entire body increased significantly from 6 dpf in *AXER* −/− medaka compared with *AXER* +/+ medaka (Fig 2C). Thus, protective UPR signaling was switched to cell death signaling during 4–6 dpf in *AXER* −/− medaka, leading to the death of all *AXER* −/− medaka by 12 dpf (Fig 2D).

## Occurrence of heart failure in *AXER*-KO medaka

We found that blood flow in the caudal vein above the yolk in *AXER* −/− medaka was markedly slowed from 7 dpf, almost completely stopped at 8 dpf, and completely stopped at 9 dpf (Figs 3A and S2) before death, which was judged because of cardiac arrest. Accordingly, we focused on the phenotypes of hearts.

RNA-seq analysis revealed that expression levels of various marker genes in hearts were comparable between *AXER* +/+ and *AXER* −/− medaka at 5 dpf (Fig 3B), namely, genes involved in heart looping, *TBX20* encoding a T-box family member, which is expressed in the cardiac crescent, then in the endocardium and myocardium of the linear and looped heart tube (Cai et al, 2005), *GATA4* encoding a zinc finger–containing transcription factor, which is a critical regulator of cardiac gene expression for cardiomyocyte differentiation (Molkentin et al, 1997; Watt et al, 2004), and *ISL1* encoding a homeodomain transcription factor, which is essential for the development of both arterial and venous poles (Cai et al, 2003; Pandur et al, 2013); ventricle marker genes, *HAND1* encoding a member of the basic helix–loop–helix (bHLH) transcription factor family, which is asymmetrically expressed in the developing ventricle chambers (Vincentz et al, 2017), and *HEY2* encoding a hairy-related bHLH transcription factor, which acts for the differentiation of pluripotent stem cells into ventricular myocardial cells (Koibuchi & Chin, 2007); atrial marker genes, *HEY1* encoding a member of the hairy and enhancer of split-related family of bHLH-type transcription factors, which works as a downstream effector of Notch signaling required for cardiovascular development (Fischer et al, 2007), and *NPPA* encoding atrial natriuretic peptide, an early and specific marker for differentiation of the myocardium, which is implicated in the control of extracellular fluid volume and electrolyte homeostasis (Houweling et al, 2005); and sarcomeric genes, *MYH7* encoding β-myosin heavy chain, which is found in heart muscle and slow-twitch type 1 skeletal muscle fibers (Epstein et al, 1992), *MYBPC3* encoding a cardiac isoform of myosin-binding protein C expressed exclusively

in cardiac muscle, which is associated with the structure of the sarcomere (basic unit of muscle contraction) (Harris et al, 2002), and *TNNT2* encoding cardiac troponin T, one of three proteins that make up the troponin protein complex, which is located on the thin filament of striated muscles and regulates muscle contraction (Watkins et al, 1995).

During the differentiation process, termed cardiac looping, the linear heart tube forms at 3 dpf, and rotates and bends into an S-shaped loop at 4 dpf, which in turn results in the formation of segmented chambers, the ventricle and atrium, at 5 dpf. This process appeared to occur normally in *AXER* −/− medaka (Fig 3C). We then observed abnormal phenotypes in hearts of *AXER* −/− medaka from 6 dpf. Note that Venus expression from the VC3Ai reporter in the ventricle increased significantly in *AXER* −/− medaka from 4 dpf compared with *AXER* +/+ medaka, and the difference continued to increase toward 7 dpf (Fig 4A). The length of the ventricle became significantly shorter in *AXER* −/− medaka than in *AXER* +/+ medaka from 6 dpf at both diastole and systole, and the difference continued to increase toward 8 dpf (Fig 4B). In contrast, although the length of the atrium became gradually shorter during embryonic development (6–8 dpf) in *AXER* +/+ medaka, it became significantly longer in *AXER* −/− medaka than in *AXER* +/+ medaka from 6 dpf, and the difference continued to increase toward 8 dpf (Fig 5A). The heart in *AXER* −/− medaka exhibited significantly increased beats per minute compared with *AXER* +/+ medaka from 6 dpf (Fig 5B). Furthermore, pericardial fluid surrounding the heart was markedly increased in *AXER* −/− medaka compared with *AXER* +/+ medaka (Fig 5C). These results clearly indicate the occurrence of heart failure in *AXER* −/− medaka.

## Rescue of heart failure in *AXER*-KO medaka by constitutive activation of XBP1 but not ATF6α

We then examined whether constitutive activation of a particular pathway of the UPR could rescue the heart failure observed in *AXER* −/− medaka. Previously, using a genome-editing technique, we produced mutant medaka designated *XBP1(S^C)*, in which the active (spliced) form of XBP1, XBP1(S), is constitutively expressed from the 26-nucleotide-intron-less *XBP1* locus (Ishikawa et al, 2017). In the present study, to produce mutant medaka designated *ATF6α(N^C)*, in which the active (nuclear) form of ATF6α, ATF6α(N), is constitutively expressed, we deleted the DNA region corresponding to the luminal and most transmembrane regions from the *ATF6α* locus using the CRISPR/Cas9 technology (Fig S3A(a)), resulting in the production of ATF6α(N^C) of 377 aa containing the basic leucine zipper (bZIP) domain (Fig S3A(b)). The expected deletion of the DNA region was confirmed by sequencing and genomic PCR (Fig S3B(a and b)). *ATF6α N^C*/+ medaka hatched normally (Fig S3C) and exhibited a normal phenotype, including a normal tail length

AXER protein is schematically shown in comparison with the WT AXER protein containing eight transmembrane domains at right. The red star denotes the stop codon for Δ2. **(D)** Hatching rates of *AXER* +/+, +/−, and −/− medaka obtained by incrossing male and female *AXER* +/− medaka were determined. Gray and black bars indicate hatched and unhatched medaka, respectively. **(E)** *AXER* −/− medaka are compared with *AXER* +/+ medaka at birth (1 dph). Scale bar, 1 mm. **(F)** (a) Positions of tails of *AXER* +/+ and −/− medaka at 1 dph are shown in red. Scale bar, 250 μm. (b) Tail lengths were measured (n = 6 for +/+, 15 for +/−, and 5 for −/−). **(G)** Male and female *AXER* ± medaka were incrossed, and 46 resulting hatched fish were genotyped 2 mo later.
Source data are available for this figure.

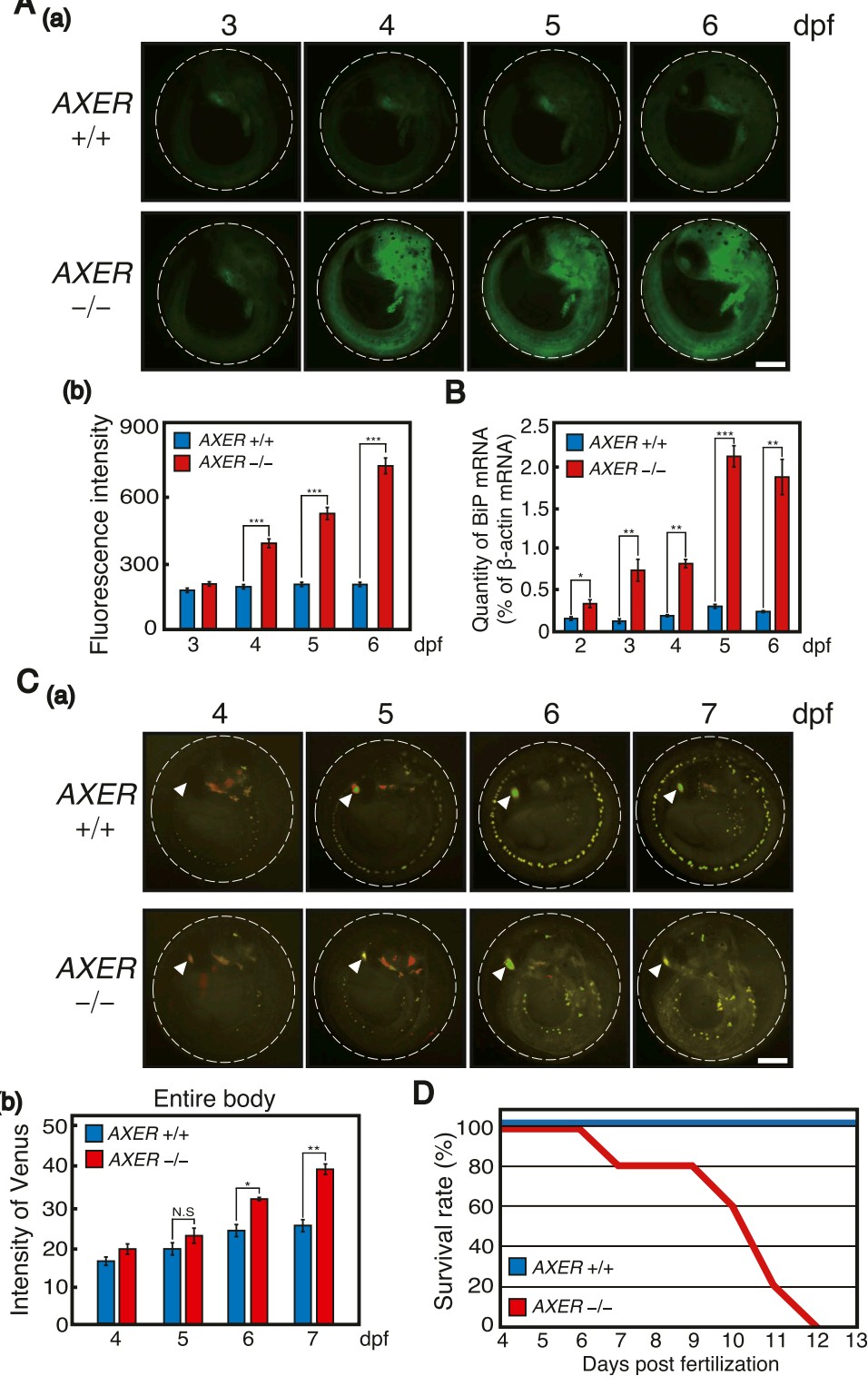

**Figure 2. Monitoring of ER stress and apoptosis in *AXER*-KO medaka.**
**(A)** Male *AXER* +/− medaka carrying the P$_{BiP}$-EGFP reporter and female *AXER* +/− medaka were crossed. (a) *AXER* +/+ and −/− medaka carrying the P$_{BiP}$-EGFP reporter were analyzed by fluorescence microscopy at 3–6 dpf. Scale bar, 250 μm. (b) Fluorescence intensities in the entire body were quantified (n = 6 for +/+ and 5 for −/−). **(B)** Quantitative RT–PCR was conducted to determine the level of endogenous BiP mRNA relative to that of β-actin mRNA in *AXER* +/+ and −/− medaka (n = 3). **(C)** Male *AXER* +/− medaka carrying the VC3Ai reporter were crossed with female *AXER* +/− medaka. (a) *AXER* +/+ and −/− medaka carrying the VC3Ai reporter were analyzed by fluorescence microscopy at 4–7 dpf. Scale bar, 250 μm. (b) Fluorescence intensities in the entire body were quantified (n = 6 for +/+ and 4 for −/−). **(D)** Survival rates of *AXER* +/+ and −/− medaka were determined (n = 15 for +/+ and 5 for −/−).

(Fig S3D). ATF6α(N$^C$) fully rescued the embryonic lethality of *ATF6α/ATF6β*-double KO medaka (Fig S3E). After crossing with WT fish carrying the P$_{BiP}$-EGFP reporter, *ATF6α N$^C$/+* medaka carrying the P$_{BiP}$-EGFP reporter exhibited more profound fluorescence in the entire body than WT medaka carrying the P$_{BiP}$-EGFP reporter (Fig S3F). RNA-seq analysis revealed that expression levels of various ER chaperones in hearts were enhanced in *ATF6α N$^C$/+* medaka compared with WT medaka at 5 dpf, as expected (Fig S3G).

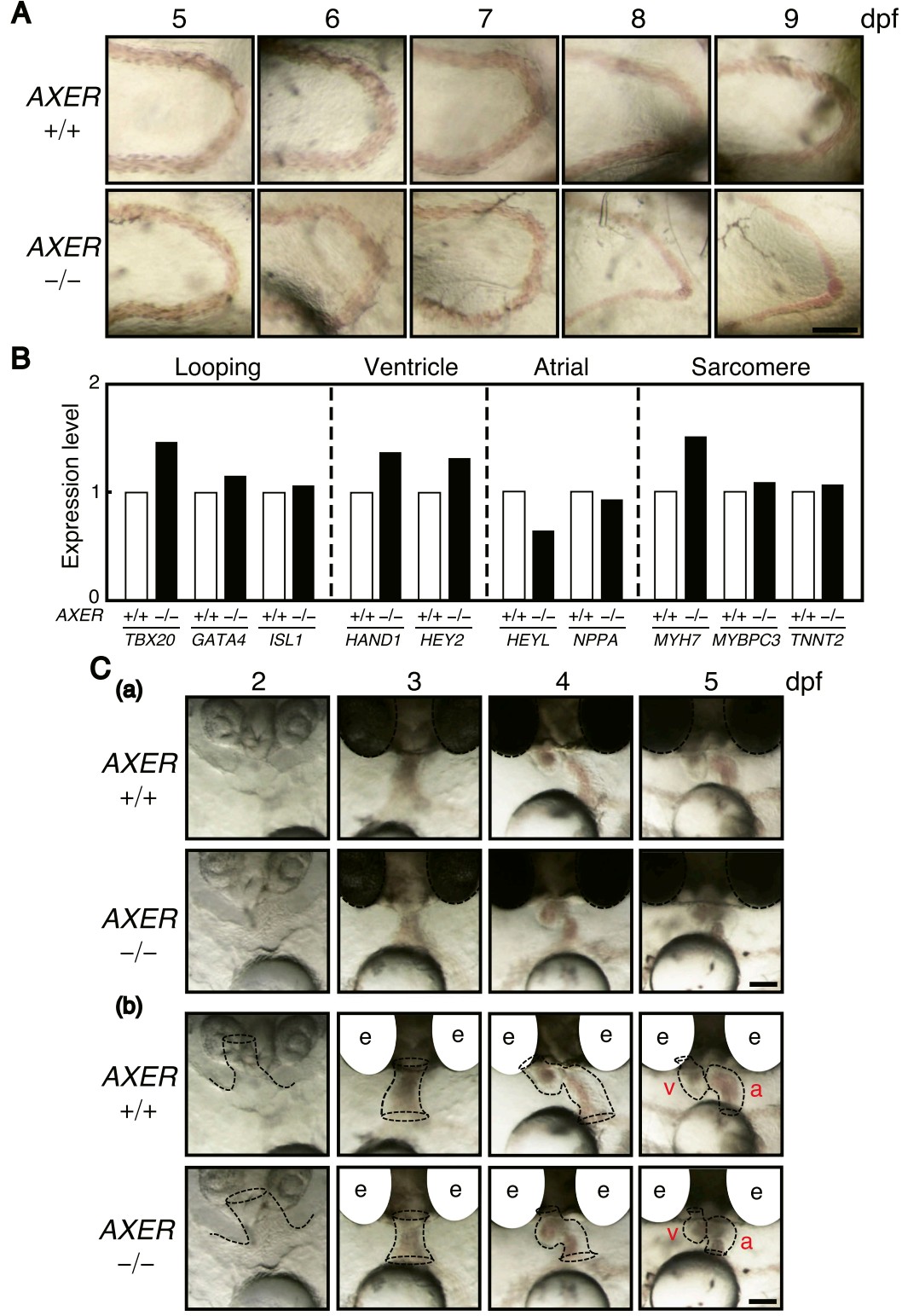

**Figure 3.   Effect of *AXER*-KO on blood flow, heart gene expression, and cardiac looping.**
**(A)** Caudal vein above the yolk in *AXER* +/+ and −/− medaka was photographed at 5–9 dpf. Scale bar, 100 μm. **(B)** Expression levels of various marker genes in hearts of *AXER* −/− medaka relative to those in hearts of *AXER* +/+ medaka at 5 dpf were determined by RNA-seq. **(C)** (a) Region in which the heart is eventually formed via cardiac looping was photographed in *AXER* +/+ and −/− medaka at 2–5 dpf. Scale bar, 100 μm. (b) Process of cardiac looping is illustrated by dashed lines. e, eye; v, ventricle; a, atrium. Scale bar, 100 μm.

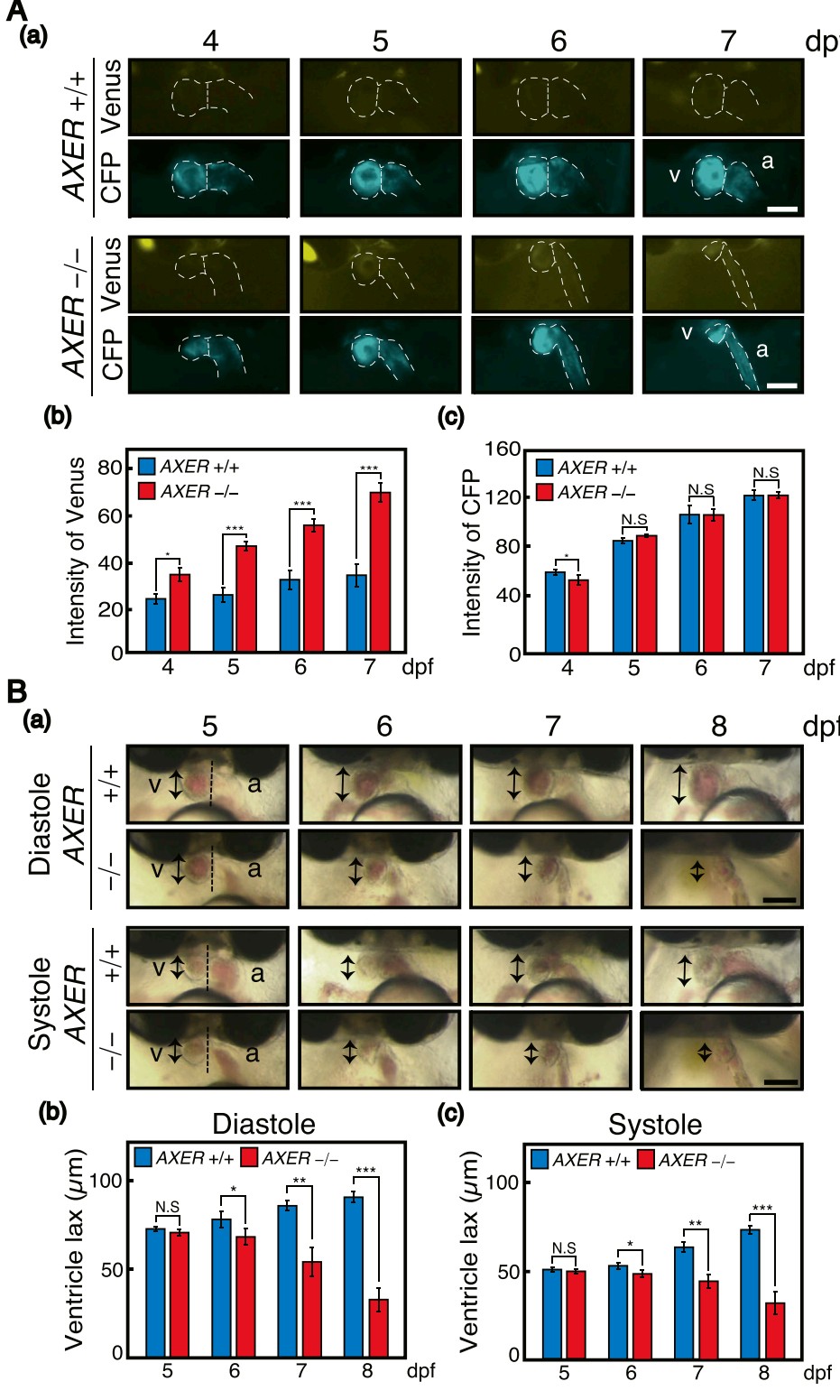

Figure 4. Heart (ventricle) failure in *AXER*-KO medaka.
**(A)** Male *AXER* +/− medaka carrying the VC3Ai reporter were crossed with female *AXER* +/− medaka. (a) Hearts of *AXER* +/+ and −/− medaka carrying the VC3Ai reporter were analyzed by fluorescence microscopy at 4–7 dpf. Scale bar, 100 $\mu m$. (b, c) Fluorescence intensities of (b) Venus and (c) CFP in the ventricle were quantified (n = 6). **(B)** Male and female *AXER* +/− medaka were incrossed. (a) Hearts of *AXER* −/− medaka were compared with those of *AXER* +/+ medaka at 5–8 dpf. v, ventricle; a, atrium. Bidirectional arrows indicate the length of the ventricle. Scale bar, 100 $\mu m$. (b, c) Ventricle lengths at (b) diastole and (c) systole were measured (n = 6).

For analysis of phenotypes, *AXER* +/− medaka were crossed with *AXER* +/+ *XBP1* $S^C$/+ medaka and *AXER* +/+ *ATF6α* $N^C$/+ medaka to obtain *AXER* +/− *XBP1* $S^C$/+ medaka and *AXER* +/− *ATF6α* $N^C$/+ medaka, respectively. Male *AXER* +/− *XBP1* $S^C$/+ medaka were crossed with female *AXER* +/− *ATF6α* $N^C$/+ medaka, and male *AXER* +/− *ATF6α* $N^C$/+ medaka were crossed with female *AXER* +/− *XBP1* $S^C$/+ medaka to obtain (1) *AXER* +/+ *ATF6α* +/+ *XBP1* +/+ medaka, (2) *AXER* +/+ *ATF6α* $N^C$/+ *XBP1* +/+ medaka, (3) *AXER* +/+ *ATF6α* +/+ *XBP1* $S^C$/+ medaka,

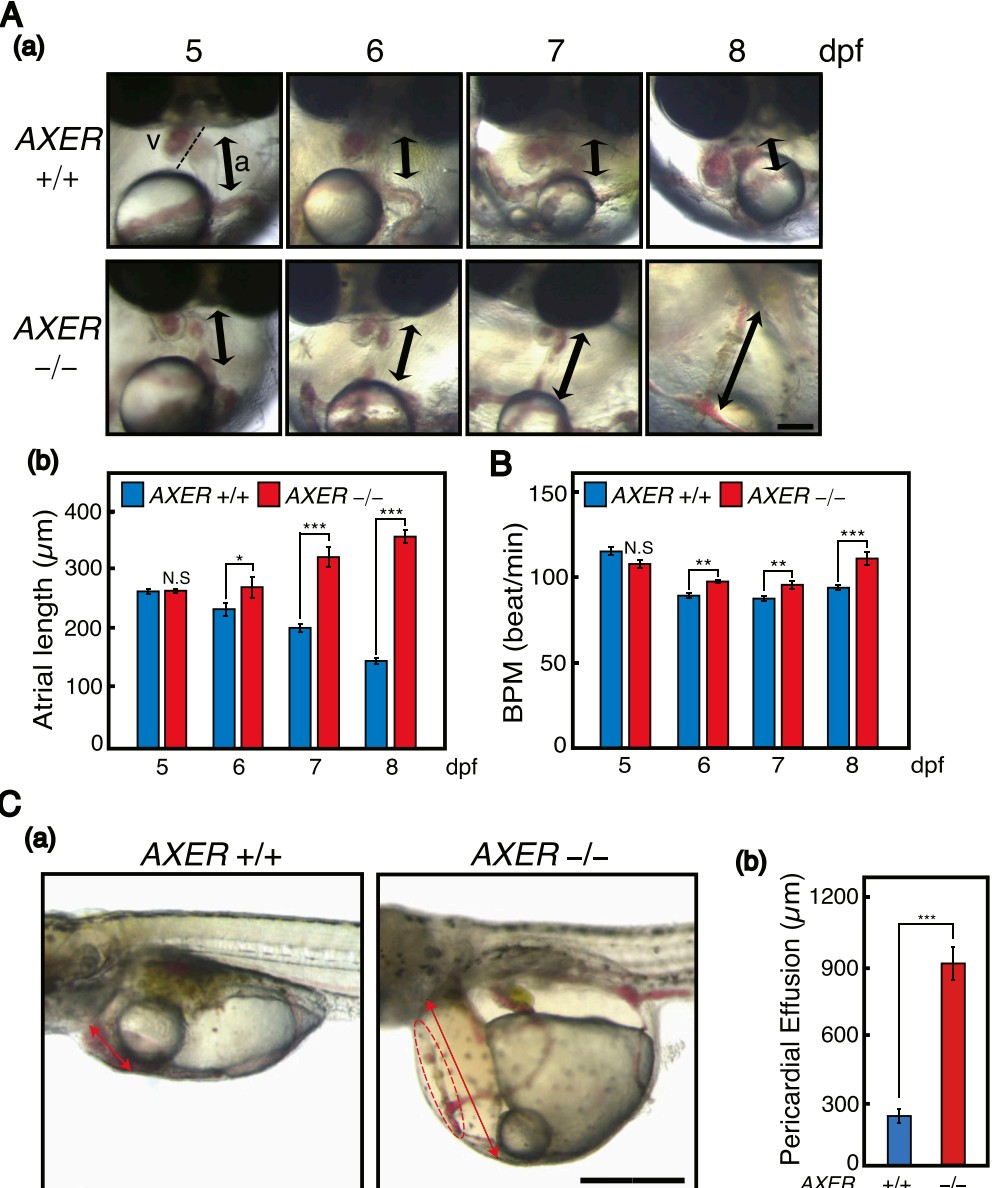

**Figure 5. Heart failure in *AXER*-KO medaka.**
**(A)** Male and female *AXER* +/− medaka were incrossed. (a) Hearts of *AXER* −/− medaka were compared with those of *AXER* +/+ medaka at 5–8 dpf. Thick bidirectional arrows indicate the length of the atrium. Scale bar, 100 μm. (b) Atrium lengths were measured (n = 10 for +/+ and 4 for −/−). **(B)** Beats per minute in hearts of *AXER* +/+ and −/− medaka at 5–8 dpf were determined (n = 10 for +/+ and 4 for −/−). **(C)** (a) Pericardial fluid surrounding the heart of *AXER* +/+ and −/− medaka was photographed at 8 dpf. Scale bar, 500 μm. The heart of *AXER* −/− medaka is enclosed within the dashed red line. (b) Lengths of pericardial effusion indicated by the red bidirectional arrow in (a) were measured (n = 8 for +/+ and 4 for −/−).

(4) *AXER* −/− ATF6α +/+ XBP1 +/+ medaka, (5) *AXER* −/− ATF6α N$^C$/+ XBP1 +/+ medaka, and (6) *AXER* −/− ATF6α +/+ XBP1 S$^C$/+ medaka (Figs 6 and 7). We found that the constitutive expression of XBP1(S) from the *XBP1(S$^C$)* locus but not that of ATF6α(N) from the *ATF6α(N$^C$)* locus significantly rescued abnormal phenotypes of *AXER* −/− medaka, namely, the poor hatching rate (Fig 6A); crooked and shorter tail (Fig 6B and C); increased fluorescence from VC3Ai in the entire body (Fig 6D) and ventricle (Fig 6E); shortened length of the ventricle at both diastole and systole (Fig S4A); elongated length of the atrium (Fig S4B); increased beats per minute (Fig S4C); and markedly increased pericardial fluid surrounding the heart (Fig 7A).

We monitored and visualized blood flow velocity in the caudal vein above the yolk at 7 dpf by tracking the movement of red blood cells. Results clearly showed that blood flow velocity was markedly slowed in *AXER* −/− medaka compared with *AXER* +/+ medaka, but returned to close to the WT level by the constitutive expression of XBP1(S) but not ATF6α(N) (Fig 7B). We also monitored and visualized blood flow in the heart by determining the intensity of a red blood cell–dependent signal in the ventricle and atrium (Fig S4D). Results again showed that blood flow was markedly slowed in *AXER* −/− medaka compared with *AXER* +/+ medaka, but returned to close to the WT level by the constitutive expression of XBP1(S) but not ATF6α(N) (Fig 7C). Accordingly, *AXER* −/− ATF6α +/+ XBP1 S$^C$/+ medaka lived 3 d longer than *AXER* −/− ATF6α +/+ XBP1 +/+ medaka or *AXER* −/− ATF6α N$^C$/+ XBP1 +/+ medaka (Fig 7D).

## RNA-seq analysis

To explain the differential effects of the constitutive expression of XBP1(S) and ATF6α(N) on the heart defects observed in *AXER* −/−

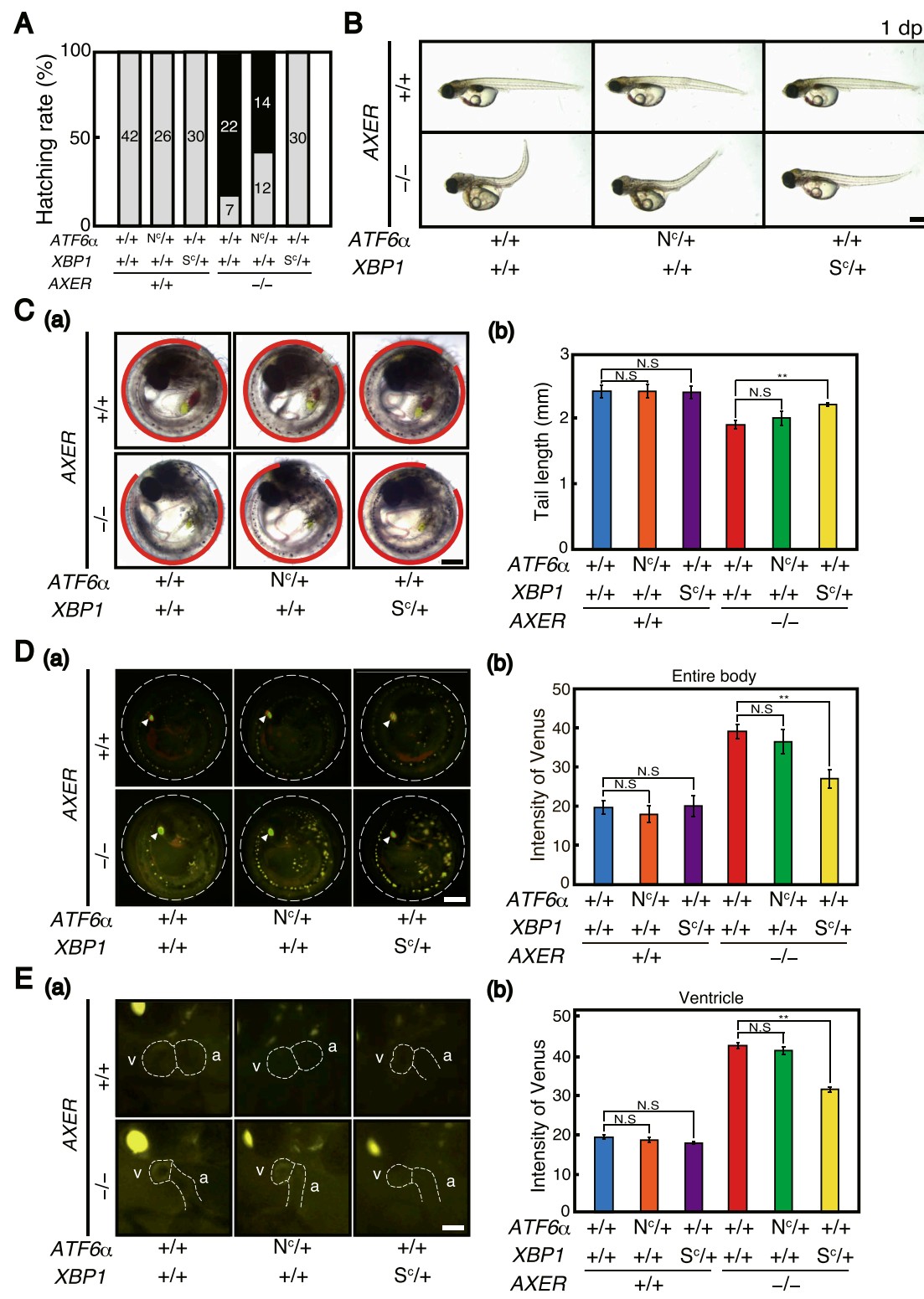

**Figure 6. Rescue of heart failure in *AXER*-KO medaka by constitutive activation of XBP1 but not ATF6α.**

Male *AXER +/− XBP1 S^C/+* medaka were crossed with female *AXER +/− ATF6α N^C/+* medaka, and male *AXER +/− ATF6α N^C/+* medaka were crossed with female *AXER +/−XBP1 S^C/+* medaka to obtain medaka of the indicated genotypes. **(A)** Hatching rates of various medaka with the indicated genotypes were determined. Gray and black bars indicate hatched and unhatched medaka, respectively. **(B)** Various medaka with the indicated genotypes were photographed at 1 dph. Scale bar, 1 mm. **(C)** (a) Positions of the tail of various medaka with the indicated genotypes at 7 dpf are shown. Scale bar, 250 μm. (b) Tail lengths were measured (n ≥ 4). **(D)** (a) Various medaka carrying the VC3Ai reporter with the indicated genotypes were analyzed by fluorescence microscopy at 7 dpf. White arrowheads denote merged fluorescence of Venus and mCherry

medaka, we conducted RNA-seq analysis on RNA samples prepared from hearts at 5 dpf of (a) *AXER +/+ ATF6α +/+ XBP1 +/+* (WT) medaka, (b) *AXER −/− ATF6α +/+ XBP1 +/+* (*AXER*-KO) medaka, (c) *AXER −/− ATF6α +/+ XBP1 S^C/+* [KO + XBP1(S)] medaka, and (d) *AXER −/− ATF6α N^C/+ XBP1 +/+* [KO + ATF6α(N)] medaka.

When the results of (a) WT medaka and (b) *AXER*-KO medaka were compared, expression levels of 378 genes were significantly altered (>1.5-fold, q-value <0.05), with 266 genes up-regulated and 112 genes down-regulated in *AXER*-KO medaka (Fig 8A). When the results of (b) KO medaka and (c) [KO + XBP1(S)] medaka or (d) [KO + ATF6α(N)] medaka were compared, expression levels of 413 and 48 genes were significantly altered (>1.5-fold, q-value <0.05) by the constitutive expression of XBP1(S) and ATF6α(N), respectively (Fig 8B), consistent with previous findings that XBP1 has much broader targets than ATF6α (Acosta-Alvear et al, 2007; Adachi et al, 2008). For example, the expression level of *LMAN1*, a known target of the IREα-XBP1 pathway, was increased in KO medaka compared with WT medaka and further increased in [KO + XBP1(S)] medaka compared with KO medaka, but not increased in [KO + ATF6α(N)] medaka compared with KO medaka (Fig 8B, right panel). On the contrary, the expression level of *HSPA5* (BiP), a known target of the ATF6 pathway, was increased in KO medaka compared with WT medaka, and further increased in [KO + ATF6α(N)] medaka compared with KO medaka, whereas *NUPRIB*, a known target of the PERK pathway, was increased in KO medaka compared with WT medaka, but not increased in [KO + ATF6α(N)] medaka or [KO + XBP1(S)] medaka compared with KO medaka, as expected (Fig 8B, right panel). Importantly, the increased the expression level of *HSPA5* (BiP) in KO medaka over WT medaka was markedly decreased in [KO + XBP1(S)] medaka. We considered that this is due to amelioration of ER stress in *AXER*-KO medaka through the constitutive expression of XBP1(S), which up-regulated 206 of 413 altered genes and thereby helped restoration of the ER homeostasis of the heart.

Then, we drew three types of comparison in a Venn diagram with the number of genes whose expression levels were significantly altered (>1.5-fold, q-value <0.05), namely, (i) comparison between WT and KO medaka, (ii) comparison between KO and [KO + XBP1(S)] medaka, and (iii) comparison between WT and [KO + XBP1(S)] medaka. This was followed by typical alteration patterns of each category with total (boxed), up-regulated (blue upward-pointing triangle), and down-regulated (red downward-pointing triangle) gene numbers in each category (Fig 8C); note that the typical alteration patterns (bar graphs) show only the case of up-regulation in KO medaka compared with WT medaka, except for the focused (see below) 77 genes. The central category (dark green), containing 13 genes, represented *LMAN1*-like XBP1 target genes, whose expression levels were significantly increased in KO medaka compared with WT medaka, and further significantly increased in [KO + XBP1(S)] medaka compared with KO medaka (compare white, blue, and purple bars, Fig S5).

Here, we considered that the category containing 77 genes (dark pink) was most important in understanding how the constitutive expression of XBP1(S) rescued the heart failure observed in KO

medaka, as their expression levels were significantly increased (69 genes) or decreased (eight genes) in KO medaka compared with WT medaka, but significantly decreased (69 genes) or increased (eight genes) in [KO + XBP1(S)] medaka compared with KO medaka. Accordingly, expression levels in WT medaka and in [KO + XBP1(S)] medaka became comparable (not significant). When the rescue rate was defined as Y/X x 100 (%), where X and Y represent the difference between KO medaka and WT medaka, and between [KO + XBP1(S)] medaka and KO medaka, respectively, all 77 genes showed more than 50% rescue rates (Fig 8D). Of note, although BiP (and calreticulin and ERdj4, both with *P* > 0.05) was categorized in this category, most ER chaperones (GRP170, GRP94, ERp72, P5, and GRP58) were categorized in the left middle (green) category containing 147 genes, whose expression levels were significantly increased in KO medaka compared with WT medaka, but the decrease in [KO + XBP1(S)] medaka compared with KO medaka was not significant enough (less than 50% rescue rates), because direct transcriptional induction of these genes by XBP1(S) (compare magenta bars with white bars) obscured the decrease mediated by amelioration of ER stress through the constitutive expression of XBP1(S) (compare blue bars with purple bars, Fig S5).

Because erythroblasts undergo enucleation only in mammals, it turned out that RNA-seq data were obtained from a mixture of RNA derived from hearts and erythrocytes. Therefore, we classified the 77 genes into five categories: (1) heart-specific expression, (2) greater than twofold expression in hearts than in erythrocytes (heart > erythrocyte expression), (3) nearly equal expression in hearts and erythrocytes (heart ~ erythrocyte expression), (4) greater than twofold expression in erythrocytes than in hearts (erythrocyte > heart expression), and (5) erythrocyte-specific expression, using "BioGPS" (http://biogps.org/#goto=welcome), a gene annotation portal (Wu et al, 2009). In Fig 9, 69 genes up-regulated in KO medaka were classified into five categories with blue, purple, and black bars indicating a fold increase in KO medaka over WT medaka, rescue rate by XBP1(S), and rescue rate by ATF6α(N), respectively. *HSPA5* encoding BiP is listed as 11th in the heart > erythrocyte category. Remarkably, the increase in all 69 genes in *AXER*-KO (blue bars, Fig 9) was well rescued by the constitutive expression of XBP1(S) (purple bars, Fig 9) but was barely rescued by the constitutive expression of ATF6α(N) (black bars, Fig 9), which was highly consistent with the results of phenotype rescue by XBP1(S) (Figs 6 and 7).

Among the up-regulated 69 genes, we discovered several interesting genes (marked with red circles in Fig 9), whose increased expression was reported to adversely affect the function of the heart (see the Discussion section for a detailed explanation). Indeed, abnormal phenotypes (ventricle became smaller; atrium did not become round and instead became longer or enlarged) were frequently observed in the heart of WT medaka at 7 dpf after microinjecting their mRNA into one-cell-stage embryos (Fig 10A).

In Fig S6A, eight genes down-regulated in KO medaka were classified into five categories (two categories with 0 genes) with red, purple, and black bars indicating a fold decrease in KO medaka over

from the eye lens. Scale bar, 250 μm. (b) Fluorescence intensities in the entire body were quantified (n ≥ 3). **(E)** (a) Hearts of various medaka carrying the VC3Ai reporter with the indicated genotypes were analyzed by fluorescence microscopy at 5 dpf. Scale bar, 100 μm. (b) Fluorescence intensities in the ventricle were quantified (n ≥ 3).

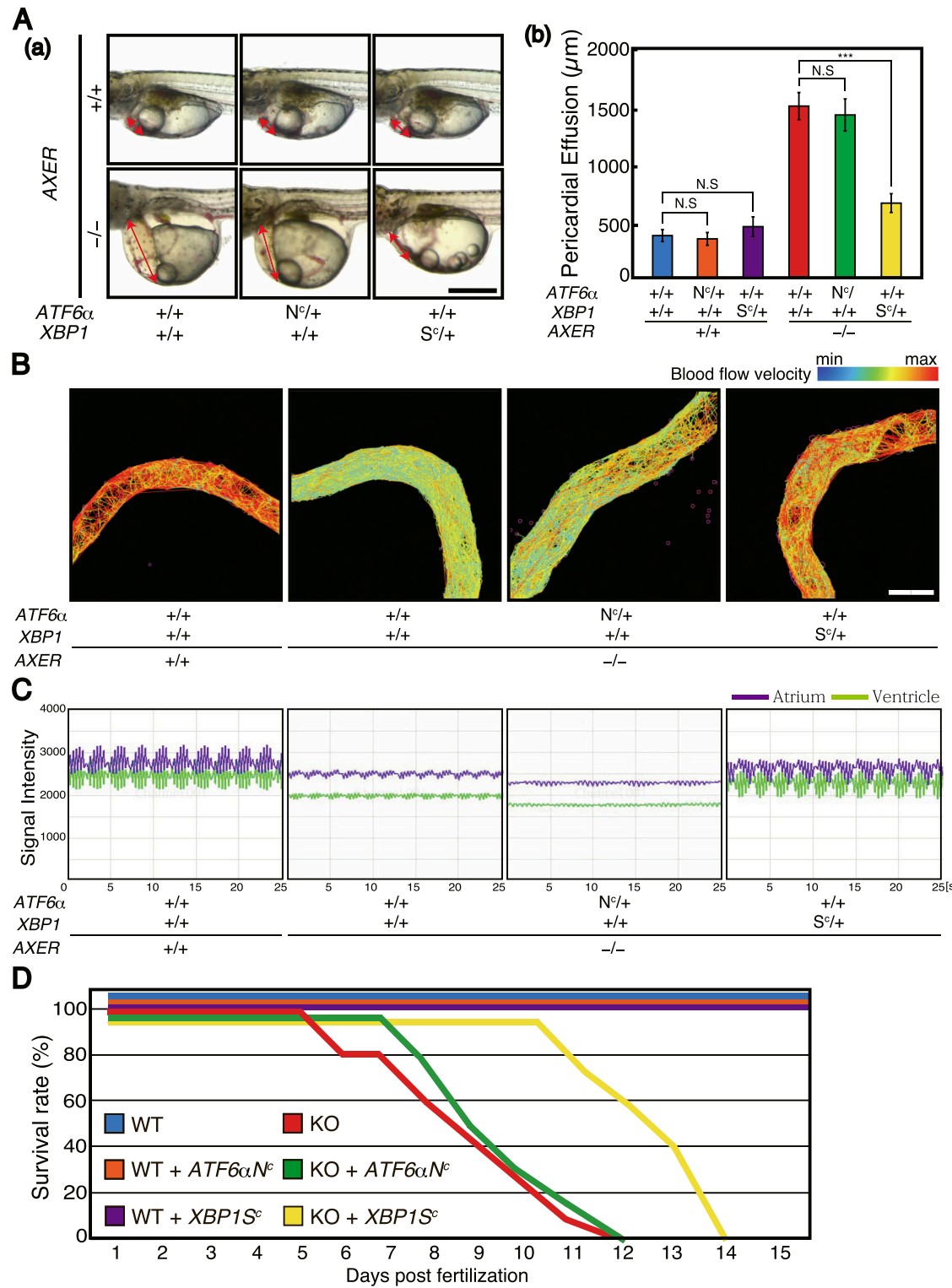

**Figure 7. Extension of lifespan of *AXER*-KO medaka by constitutive activation of XBP1 but not ATF6α.**
Male *AXER* +/− *XBP1 S^C/+* medaka were crossed with female *AXER* +/− *ATF6α N^C/+* medaka, and male *AXER* +/− *ATF6α N^C/+* medaka were crossed with female *AXER* +/− *XBP1 S^C/+* medaka to obtain medaka of the indicated genotypes. **(A)** Pericardial fluid surrounding the heart of various medaka with the indicated genotypes was photographed at 8 dpf. Scale bar, 500 μm. (b) Lengths of pericardial effusion indicated by the red bidirectional arrow in (a) were measured (n ≥ 4). **(B)** Blood flow velocity in the caudal vein of various medaka with the indicated genotypes at 7 dpf was determined by tracking and visualizing the movement of 6,000–8,000 red blood cells and is shown as a heatmap from minimum (blue) to maximum (red). Scale bar, 50 μm. **(C)** Changes in the abundance of red blood cells in the atrium (purple) and ventricle

WT medaka, rescue rate by XBP1(S), and rescue rate by ATF6α(N), respectively. In contrast to up-regulated genes, the decrease in these genes in *AXER*-KO (red bars) was rescued by the constitutive expression of ATF6α(N) (black bars) more effectively than that of XBP1(S) (purple bars), indicative of the lesser importance of these down-regulated genes in phenotype rescue by XBP1(S). Of note, we found that at least *PEPD* mRNA (first in the heart ~ erythrocyte category) contained a potential cleavage site by regulated IRE1-dependent decay of mRNAs (RIDD) by prediction using the Vienna RNA Websuite (Gruber et al, 2008) (Fig S6B), suggesting that *PEPD* mRNA was decreased in *AXER*-KO medaka heart by hyperactivated IRE1α-mediated relatively non-specific cleavage and that *PEPD* mRNA was increased by XBP1(S)-mediated amelioration of ER stress and probably by ATF6α(N)-mediated direct transcriptional induction also.

Major transcriptional targets of the UPR in maintaining the homeostasis of the ER are ER chaperones and ERAD components. In mammals, both ATF6α(N) homodimer and XBP1(S) homodimer activate transcription of ER chaperone genes via binding to cis-acting ER stress–response element with the notion that ATF6α(N) is a more potent activator than XBP1(S) (Yamamoto et al, 2007). We obtained similar results by quantitative RT–PCR in embryos at 4 dpf of WT medaka, *AXER*-KO medaka, *AXER*-KO medaka constitutively expressing ATF6α(N), and *AXER*-KO medaka constitutively expressing XBP1(S) (Fig 10B(a)). In contrast, because induction of ERAD components required both ATF6α(N) and XBP1(S) in mammals, ATF6α(N)-XBP1(S) heterodimer activates transcription of ERAD components presumably via binding to a cis-acting UPR element (Yamamoto et al, 2007, 2008). In medaka embryos, however, activation of the IRE1-XBP1 pathway appeared to be sufficient for transcriptional induction of ERAD components (Ishikawa et al, 2013). We confirmed this preferential effect of XBP1(S) on ERAD components by quantitative RT–PCR in embryos at 4 dpf of WT medaka, *AXER*-KO medaka, *AXER*-KO medaka constitutively expressing ATF6α(N), and *AXER*-KO medaka constitutively expressing XBP1(S) (Fig 10B(b)). Note that 4 dpf represents several days earlier than 7–8 dpf when we observed the rescue of abnormal phenotypes of the heart by the constitutive expression of XBP1(S) (Fig 7).

## Discussion

The heart is the first functional organ formed in the vertebrate embryo, and correct alignment of the cardiac anterior–posterior axis with the embryonic axis is essential for the establishment of correct blood flow (Kinoshita et al, 2009). We found here that *AXER*-KO burdened ER stress persistently from fertilization in medaka and caused heart failure–mediated death by 12 dpf. Although the early differentiation process of the heart, termed cardiac looping, occurring at 3–5 dpf was not affected by *AXER*-KO (Fig 3C), ER stress–induced apoptosis of the ventricle was observed from as early as 4 dpf, and altered morphology of the

ventricle (shortness, Fig 4B) and atrium (elongation, Fig 5A) began to be observed from 6 dpf in *AXER*-KO medaka. Accordingly, pericardial fluid surrounding the heart was markedly increased (Fig 5C) and blood flow was slowed down from 7 dpf in *AXER*-KO medaka (Fig 3A). Importantly, these defects were well rescued by the constitutive expression of XBP1(S) but not ATF6α(N) from fertilization (Figs 6 and 7). Therefore, we conducted RNA-seq analysis using RNA samples prepared from hearts at 5 dpf to find the cause of this phenotype. Among 23,622 genes examined, expression levels of 378 genes were found to be significantly altered (266 up-regulated and 112 down-regulated) in the heart of *AXER*-KO medaka compared with that of WT medaka (Fig 8A). We focused on the 69 up-regulated genes among them, because their changes in the expression level appeared to be associated with the phenotype (Fig 8C), and found that the changes in these genes in the heart of *AXER*-KO medaka compared with WT medaka were very well rescued by the constitutive expression of XBP1(S) but were barely rescued by the constitutive expression of ATF6α(N) (Fig 9). This was well consistent with the results of phenotype rescue by these active transcription factors (Figs 6 and 7). Among them, we discovered several interesting genes (marked with red circles in Fig 9), which explain the development of heart failure, as follows. Indeed, microinjection of their mRNA into one-cell-stage embryos of WT medaka frequently produced hearts with altered morphology at 7 dpf (Fig 10A).

(1) Heart-specific genes. (1-1) *DESMA* encoding desmin (second in the category), which is the primary intermediate filament of cardiac, skeletal, and smooth muscles (Agnetti et al, 2021). The desmin network, which connects the Z-disks in adjacent myofibrils and the myofibrils to the nuclear envelope and sarcolemma, is critical for the structural integrity of cardiomyocytes, and changes in desmin filaments, including an increased level of desmin, have been reported in hypertrophic and failing hearts (Sheng et al, 2016). (1-2) *MTHFD2* encoding mitochondrial methylenetetrahydrofolate dehydrogenase/cyclohydrolase (third in the category). MTHFD2-controlled amino acid metabolism, namely, generation of glycine from serine and ultimate de novo synthesis of purines, maintains endothelial ATP levels and is essential for angiogenesis. The expression of *MTHFD2* mRNA is reported to be elevated in cardiovascular disease (Hitzel et al, 2018); note that *PHGDH* mRNA encoding cytosolic phosphoglycerate dehydrogenase (second from the bottom in the heart > erythrocyte category), which participates in MTHFD2-controlled amino acid metabolism, is also reported to be increased in cardiovascular disease (Hitzel et al, 2018).

(2) heart > erythrocyte genes. (2-1) *SOCS3* encoding a suppressor of cytokine signaling 3 (second in the category). SOCS3 is an intrinsic negative feedback regulator of the potently cardioprotective JAK-STAT signaling pathway. Based on results in cardiac-specific *SOCS3*-KO mice, SOCS3 was suggested to represent a key factor that exacerbates the development of

(green) of various medaka with the indicated genotypes were determined at 7 dpf. Each single peak (50 peaks in 25 s) represents the consequence of one contraction.
**(D)** Survival rates of various medaka with the indicated genotypes were determined (n ≥ 7).

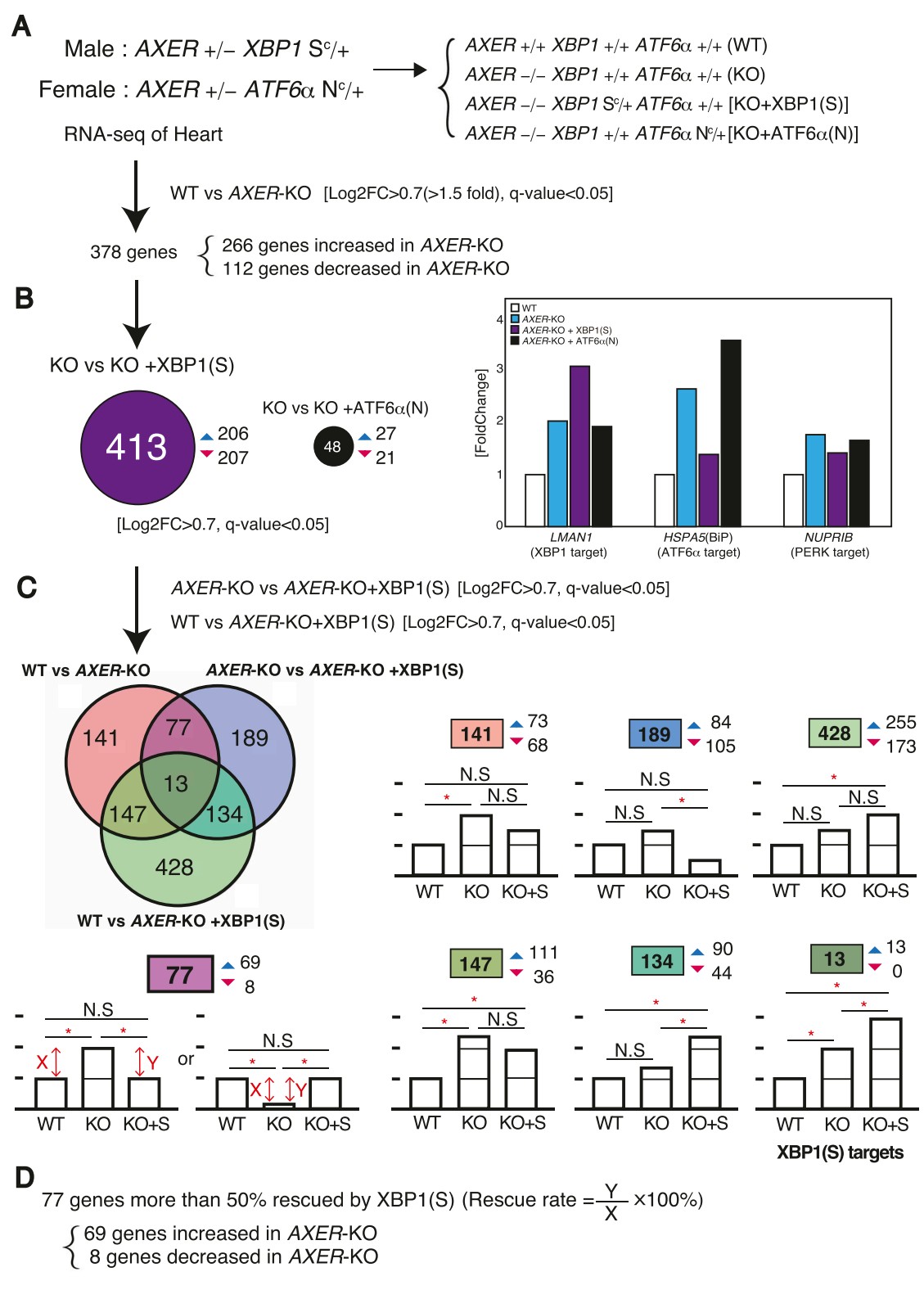

**Figure 8. RNA-seq analysis of hearts of *AXER* +/+ and −/− medaka with or without *XBP1(S^C)* or *ATF6(N^C)*.**
**(A)** Procedures of RNA-seq analysis are shown. **(B)** Differential effects of the constitutive expression of XBP1(S) and ATF6α(N) in *AXER*-KO medaka on heart gene expression in total (left) or on the expression of *LMAN1* (XBP1 target), *HSPA5* (ATF6α target), and *NUPRIB* (PERK target) (right) are shown. **(C)** Effect of the constitutive expression of XBP1(S) on *AXER* +/+ and −/− medaka is described in a Venn diagram with the number of genes whose expression levels were significantly altered (>1.5-fold, q-value <0.05), followed at right by typical alteration patterns of each category (up-regulation pattern only except for the dark pink category) together with total (boxed), up-regulated (blue upward-pointing triangle), and down-regulated (red downward-pointing triangle) gene numbers in each category. **(C, D)** All 77 genes in the dark pink category in (C) showed greater than 50% rescue rates.

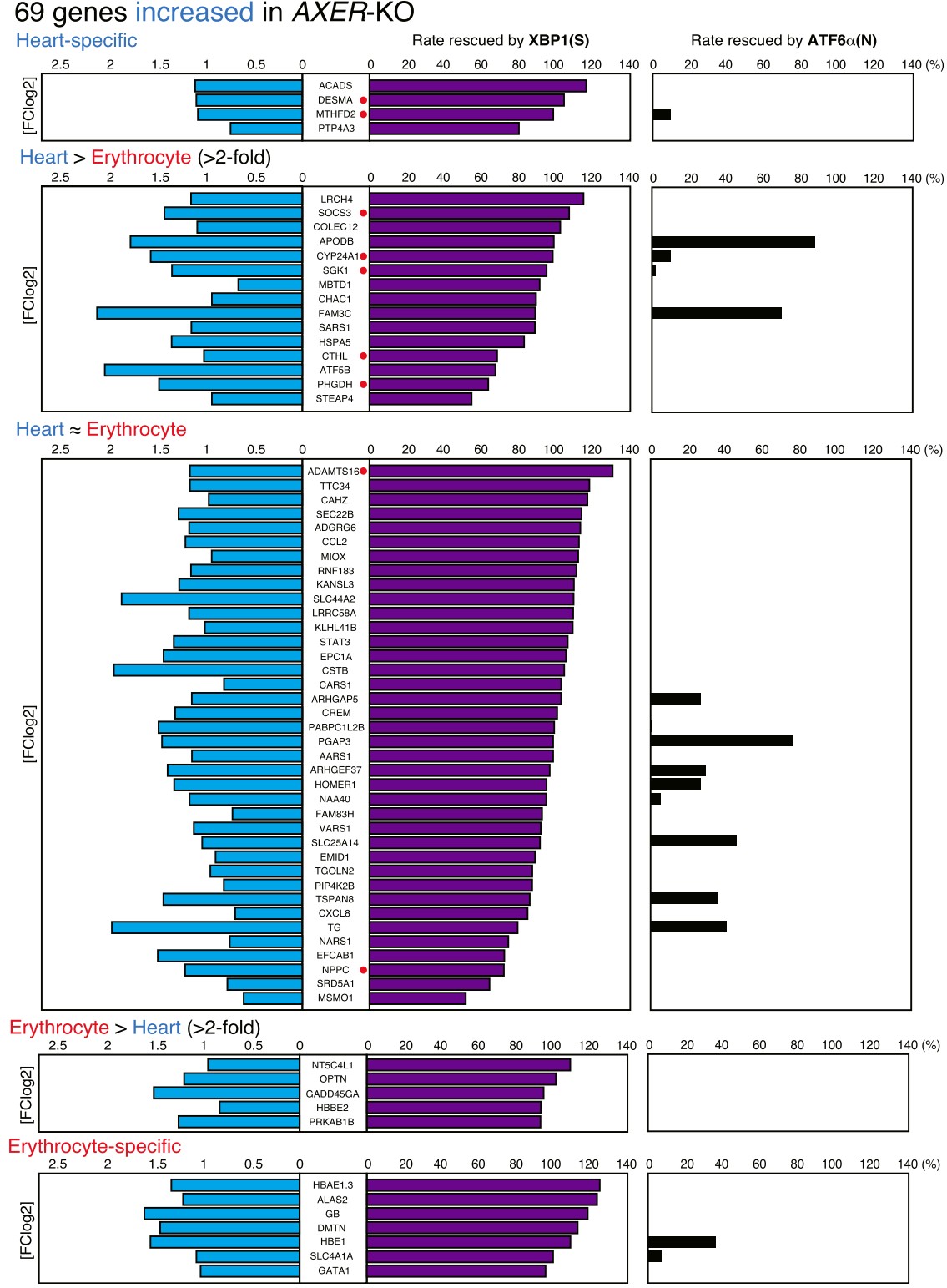

**Figure 9.  Effects of constitutive activation of XBP1 and ATF6α on 69 up-regulated genes.**
69 genes increased in *AXER*-KO medaka compared with WT medaka are shown with a fold increase in *AXER*-KO (blue bars), rescue rates by XBP1(S) (purple bars), and rescue rates by ATF6α(N) (black bars) after classification into five categories. Genes with a red circle denote those whose excess expression is known to adversely affect the function of the heart.

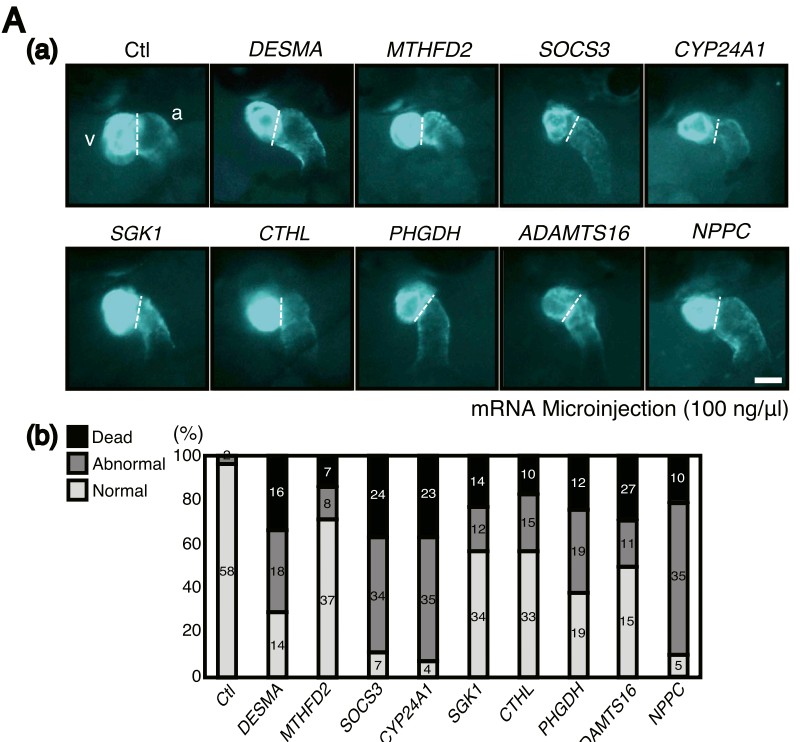

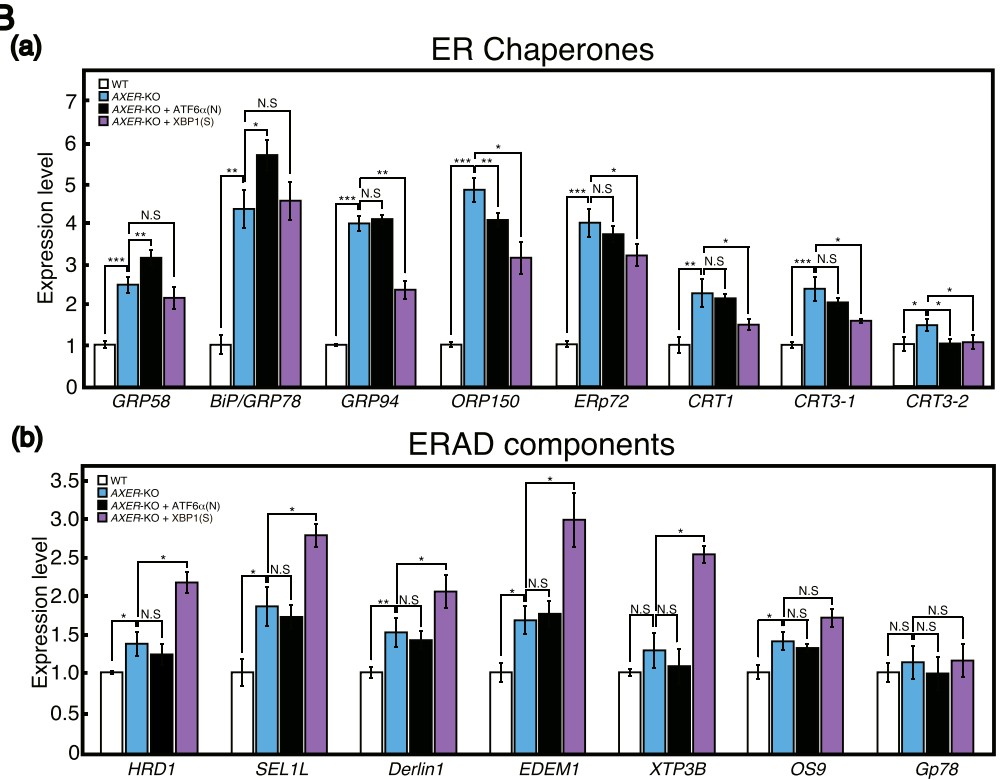

**Figure 10. Effect of microinjecting various mRNA on the phenotype of the heart, as well as the effect of constitutive activation of ATF6α and XBP1 on the levels of mRNA encoding various ER chaperones and ERAD components.**
**(A)** (a) Various mRNA was microinjected as indicated into one-cell-stage embryos of WT medaka carrying the P_cmlc2-tagCFP at the concentration of 100 ng/μl, and their hearts were observed at 7 dpf. v, ventricle; a, atrium. Scale bar, 100 μm. (b) Numbers of fish with a normal heart, fish with an abnormal heart, and dead fish were counted. **(B)** Quantitative RT–PCR was conducted to determine the levels of mRNA encoding various (a) ER chaperones and (b) ERAD components in hearts of WT medaka, *AXER*-KO medaka, *AXER*-KO medaka constitutively expressing ATF6α(N), and *AXER*-KO medaka constitutively expressing XBP1(S) at 4 dpf (n = 3). The level of each mRNA in WT medaka is set as 1.

myocardial ischemia/reperfusion injury (Nagata et al, 2015). It was also reported that plasma SOCS3 was significantly increased in acute myocardial infarction patients and that acute myocardial infarction patients with a higher plasma SOCS3 level were at higher risk for major adverse cardiac events and worse 5-yr overall survival (Xia et al, 2017). (2-2) *CYP24A1*

encoding vitamin D-24-hydroxylase, one of the enzymes responsible for vitamin D metabolism (fifth in the category). Transgenic rats constitutively expressing CYP24 showed atherosclerotic lesions in the aorta, which greatly progressed with high-fat and high-cholesterol feeding (Kasuga et al, 2002). (2-3) *SGK1* encoding serum- and glucocorticoid-regulated kinase-1 (sixth in the category). It was shown that cardiac SGK1 is activated in human and murine heart failure and that cardiac-specific activation of SGK1 in mice increased mortality, cardiac dysfunction, and ventricular arrhythmias (Das et al, 2012). (2-4) *CTHL (CSE)* encoding cystathionine gamma-lyase (fourth from the bottom in the category) (Nandi & Mishra, 2017). Cystathionine beta-synthase and CSE produce hydrogen sulfide ($H_2S$), a cardioprotective gas, from homocysteine. It was shown that the overexpression of CSE inhibited cell proliferation in HEK293 cells (Yang et al, 2004) and inhibited cell growth and stimulated apoptosis in human aorta smooth muscle cells (Yang et al, 2006).

(3)   Heart ~ erythrocyte gene. (3-1) *ADAMTS16* encoding a member of the ADAMTS superfamily of extracellular protease enzymes involved in extracellular matrix degradation and remodeling (first in the category). It was shown that the overexpression of ADAMTS16 promoted cardiac fibrosis, cardiac hypertrophy, and heart failure by facilitating cardiac fibroblast activation (Yao et al, 2020). (3-2) *NPPC* encoding a preproprotein of 126 aa that is proteolytically processed to generate C-type natriuretic peptides (CNPs) of 53 and 22 aa (third from the bottom in the category). CNPs produced by endothelial cells, cardiomyocytes, and fibroblasts play a fundamental role in cardiovascular homeostasis (Nakagawa & Nishikimi, 2022). It was shown that CNP is produced by the heart in patients with chronic heart failure (Kalra et al, 2003). Note that *NPPA* encoding natriuretic peptide A (ANP) and *NPPA*-like *CNP-3* were not transcriptionally induced in *AXER*-KO medaka (Fig S4E). *NPPB*, which encodes natriuretic peptide B (BNP), is not found in the medaka genome.

Although IRE1α-XBP1, PERK, and ATF6α/β are ubiquitously expressed, it is well known that their respective KO produces a tissue-specific phenotype, namely, failure in liver development in *XBP1*-KO mice (Reimold et al, 2000) and *XBP1*-KO medaka (Ishikawa et al, 2017), pancreatic β-cell apoptosis in *PERK*-KO mice (Harding et al, 2001), and failure in notochord development in *ATF6α/β*-double KO medaka (Ishikawa et al, 2013). This specificity indicates tissue-specific reliance on a particular pathway of the UPR to maintain the homeostasis of the ER: pancreatic β-cells rely on PERK-mediated translational control; liver hepatocytes rely on XBP1-mediated induction of various gene products; and notochord cells rely on ATF6α/β-mediated induction of ER chaperones.

In this report, we demonstrated a tissue-specific differential and protective effect of UPR signaling in a novel way using the genetically engineered medaka strains *XBP1 S$^C$/+* and *ATF6α N$^C$/+*. These have not yet been produced in mice. Persistent ER stress–induced heart failure in *AXER*-KO medaka was well rescued by the constitutive expression of XBP1(S) but not of ATF6α(N) (Figs 6 and 7), allowing *AXER*-KO *XBP1 S$^C$/+* medaka to live 3 d longer after hatching

(Fig 7D). This difference is likely ascribable to the much broader range of target genes of XBP1(S) than ATF6α(N) (Fig 8B), consistent with previous findings (Acosta-Alvear et al, 2007; Adachi et al, 2008), which successfully diminished ER stress evoked in the heart of *AXER*-KO medaka, as evidenced by RNA-seq analysis (Fig 9). In particular, transcription of various ERAD components was induced by XBP1(S) but not by ATF6α(N) (Fig 10B). Critically, luminal ATP is consumed by ER chaperones but not by ERAD, because ATP-dependent ubiquitination and degradation by the proteasome occur in the cytoplasm (Depaoli et al, 2019). Thus, ER stress evoked by the decrease in the luminal ATP level is ameliorated by ERAD of misfolded proteins accumulated in the ER, the capacity of which is enhanced by transcriptional induction of ERAD components by the IRE1-XBP1 pathway.

We will extend this type of analysis toward the brain and liver in *AXER*-KO medaka to further substantiate the tissue-specific differential effects of UPR signaling.

# Materials and Methods

## Statistics

Statistical analysis was conducted using a *t* test, with probability expressed as *$P < 0.05$, **$P < 0.01$, and ***$P < 0.001$ for all figures.

## Fish

Medaka southern strain cab was used as WT fish. Fish were maintained in a recirculating system with a 14:10-h light:dark cycle at 27.5°C. All experiments were performed in accordance with the guidelines and regulations established by the Animal Research Committee of Kyoto University (approval number: H2819). Imaging of EGFP, Venus, tagCFP, and mCherry was performed under a fluorescence stereomicroscope (Leica M205FA) using a GFP3 filter (470/40-nm excitation filter and 525/50-nm barrier filter), a YFP filter (500/20-nm excitation filter and 535/30-nm barrier filter), a CFP filter (436/20-nm excitation filter and 480/40-nm barrier filter), and a DsRed2 filter (545/30-nm excitation filter and 620/60-nm barrier filter), respectively, with a camera (Leica DFX310FX) and acquisition software (Leica LAS X). A strain carrying the P$_{BiP}$-EGFP reporter was described previously in Ishikawa et al (2011). A strain carrying the P$_{cmlc2}$-tagCFP was described previously in Ishikawa et al (2018). *ATF6α +/− and ATF6β +/−* medaka were described previously in Ishikawa et al (2013).

## Construction of plasmids

Recombinant DNA techniques were performed according to standard procedures (Sambrook et al, 1989), and the integrity of all constructed plasmids was confirmed by extensive sequencing analyses. An advanced attB-targeting vector contains medaka ubiquitin promoter–Venus–polyadenylation site (pA) and P$_{zcmlc2}$-tagCFP-pA (Ishikawa et al, 2018). The medaka ubiquitin promoter was replaced with the medaka β-actin promoter, which was obtained by PCR-mediated amplification of

the medaka genome, to create pattB-P$_{actin}$-Venus-pA-P$_{zcmlc2}$-tagCFP-pA. The VC3Ai sequence was amplified from the pCDH-puromycin-CMV-VC3Ai vector (Addgene) using Prime STAR and a pair of primers 5′-CACCATGGCCATGTACCCCTAC-GACGTGC-3′ containing the NcoI site (underlined) and 5′-GTCGCGGCCGCTTACAGGTCCTCCTCGCTG-3′ containing the NotI site (underlined) and then digested with NcoI and NotI. The resulting VC3Ai-containing fragment was used to replace the Venus sequence in pattB-P$_{actin}$-Venus-pA-P$_{zcmlc2}$-tagCFP-pA, after digestion with NcoI and NotI, to obtain pattB-P$_{actin}$-VC3Ai-pA-P$_{zcmlc2}$-tagCFP-pA.

### TALEN and phiC31 integrase methods

To construct TALEN-L and TALEN-R plasmids, TAL repeats were assembled by the modified Golden Gate assembly method (Sakuma et al, 2013). TALEN plasmids and the phiC31 integrase expression plasmid (Ishikawa et al, 2018) were linearized with NotI, purified by phenol/chloroform extraction, and used as a template to synthesize capped mRNAs using the mMESSAGE mMACHINE SP6 kit (Life Technologies) followed by purification with RNeasy MinElute (QIAGEN). Synthesized RNAs were micro-injected as described previously in Ishikawa et al (2011) into one-cell-stage embryos at the concentration of 50 ng/μl for TALEN-L and TALEN-R and 100 ng/μl for the phiC31 integrase expression plasmid.

### Genotyping

Embryos or hatched fish were suspended in 50 μl of lysis buffer (10 mM NaOH and 0.2 mM EDTA), boiled for 10 min, and then neutralized by the addition of 50 μl of 40 mM Tris–HCl, pH 8.0. The DNA fragment containing a part of AXER, XBP1, or ATF6α was amplified by PCR directly from lysates using the following primers: 5′-CAAGCGAGCGCCATTTCCAG-3′ and 5′-GAATGTAAACA-AACCGTCGAGG-3′ for AXER; 5′-GACAGAAAATGAGGAACTGAGACAGA-GAC-3′ and 5′-GACTTGAGAAACAGCTCTGGGTCAAGGAT-3′ for XBP1S$^C$; and 5′-CAGCAGCGCATGATAAAGAA-3′ (Exon9-Fw), 5′-GATCGACTGT-GAGGTCACCG-3′ (Exon16-Fw), and 5′-AGGGAAAAGTCAGAGCTGCC-3′ (Exon16-Rv) for ATF6α N$^C$. Amplified PCR fragments were subjected to digestion with a restriction enzyme in the case of AXER with BsmAI and XBP1 S$^C$ with AflIII, and then electrophoresed.

### Quantitative RT–PCR

Total RNA was extracted from embryos or hearts at the indicated dpf by the acid guanidinium/phenol/chloroform method using Isogen (Nippon Gene). Quantitative RT–PCR analysis was carried out as described previously in Ishikawa et al (2013) using the SYBR Green method (Applied Biosystems) and a pair of primers (Fw and Rv) whose names and sequences are described in Table S1.

### Quantification of blood flow in the heart and caudal vein

Blood flow in the heart and in the caudal vein above the yolk was monitored by video recording for 25 and 10 s, respectively, using a stereomicroscope (Leica M205FA) with a camera (Leica

DFX310FX). To analyze blood flow in the heart, the abundance of red blood cells in a monitoring area set in the ventricle and atrium was converted to signal intensity in inverse proportion using acquisition software (Leica LAS X). To analyze blood flow in the caudal vein, images were changed to eight bits (black and white) and run by "stack different," a plug-in tool of ImageJ2 (Fuji). Blood flow velocity was determined by tracking the movement of 6,000–8,000 red blood cells using "Track Mate" (Tinevez et al, 2017), a plug-in tool of ImageJ2, and an algorithm called "Kalman tracker."

### RNA-seq analysis

Total RNA prepared from hearts of various genotypes at 5 dpf was subjected to mRNA purification and subsequent RNA-seq analysis. RNA-seq was conducted according to the Lasy-Seq ver. 1.1 protocol (https://sites.google.com/view/lasy-seq/) (Kamitani et al, 2019; Kashima et al, 2021). Briefly, 18 ng of total RNA was reverse-transcribed with a RT primer and SuperScript IV Reverse Transcriptase (Thermo Fisher Scientific). Then, all RT mixtures of samples were pooled and purified with an equal volume of AMPure XP beads (Beckman Coulter) according to the manufacturer's manual. Second-strand synthesis was conducted on the pooled samples with RNase H (5 U/μl, Enzymatics) and DNA polymerase I (10 U/μl; Enzymatics). To avoid the carryover of a large amount of rRNAs, RNase treatment was conducted on the mixture with RNase T1 (Thermo Fisher Scientific). Then, purification was conducted with a 0.8× volume of AMPure XP beads. Fragmentation, end-repair, and A-tailing were conducted with 5× WGS Fragmentation Mix (Enzymatics). The Adapter for Lasy-Seq was ligated with 5× Ligation Mix (Enzymatics). The adapter-ligated DNA was purified with a 0.8× volume of AMPure XP beads, twice. After optimization of the PCR cycle for library amplification with quantitative RT–PCR using EvaGreen, 20× in water (Biotium), and QuantStudio 5 Real-Time PCR System (Applied Biosystems), the library was amplified with KAPA HiFi HotStart ReadyMix (KAPA BIOSYSTEMS) using the ProFlex PCR System (Applied Biosystems). The amplified library was purified with an equal volume of AMPure XP beads. One microliter of the library was used for electrophoresis using a Bioanalyzer 2100 with an Agilent High Sensitivity DNA kit (Agilent Technologies) to check quality. Sequencing of 150-bp paired-ends using HiSeq X Ten (Illumina) was carried out.

### Mapping and gene quantification

Read 1 reads were processed with fastp (version 0.21.0) (Chen et al, 2018) using the following parameters: −trim_poly_x -w 20 −adapter_sequence=AGATCGGAAGAGCACACGTCTGAACTCCAGTCA −adapter_sequence_r2=AGATCGGAAGAGCGTCGTGTAGGGAAAGA-GTGT -l 31. The trimmed reads were then mapped to the mouse reference sequences of Mus_musculus.GRCm38.cdna.all.fa, using BWA-MEM (version 0.7.17-r1188) (Li & Durbin, 2009) with default parameters. The read count for each gene was calculated with salmon using -l IU, which specifies library type (version 0.12.0) (Patro et al, 2017).

**mRNA microinjection**

cDNA of medaka *DESMA*, *MTHFD2*, *SOCS3*, *CYP24A1*, *SGK1*, *CTHL*, *PHGDH*, *ADAMTS16*, and *NPPC* genes was obtained by PCR-mediated amplification of a cDNA library constructed using mRNA of medaka at 1 dph with designed primers based on information on the Ensembl genome browser. The 5′-capped mRNA was transcribed in vitro from each cDNA by SP6 RNA polymerase using a mMESSAGE mMACHINE kit (Ambion) and then microinjected into one-cell-stage embryos at the concentration of 100 ng/μl, as described previously in Ishikawa et al (2011). EGFP mRNA was transcribed and micro-injected as a control.

# Data Availability

The data of RNA-seq analysis have been deposited in NCBI's Gene Expression Omnibus (Edgar et al, 2002) and are accessible through the GEO Series accession number GSE215040.

# Supplementary Information

# Acknowledgements

We thank Ms. Kaoru Miyagawa and Ms. Makiko Sawada for their technical and secretarial assistance. This work was financially supported in part by AMED-CREST, Japan (23gm1410005 to K Mori), and Hirose Foundation (to B Jin).

## Author Contributions

B Jin: investigation.
T Ishikawa: investigation.
M Kashima: investigation.
R Komura: investigation.
H Hirata: supervision and investigation.
T Okada: investigation.
K Mori: conceptualization, supervision, funding acquisition, and writing—original draft, review, and editing.

## Conflict of Interest Statement

The authors declare that they have no conflict of interest.

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
