## [Reviewer comments · Life Science Alliance]

Life Science Alliance

Activation of XBP1 but not ATF6a Rescues Heart Failure Induced by Persistent ER Stress in MedakaFish

Byungseok Jin, Tokiro Ishikawa, Makoto Kashima, Rei Komura, Hiromi Hirata, Tetsuya Okada, and Kazutoshi Mori
DOI: <https://doi.org/10.26508/lsa.202201771>

Corresponding author(s): Kazutoshi Mori, Kyoto University

Review Timeline:

Submission Date:	2022-10-16
Editorial Decision:	2022-11-28
Revision Received:	2023-04-21
Editorial Decision:	2023-04-24
Revision Received:	2023-04-26
Accepted:	2023-04-26

Transaction Report:

November 28, 2022

Re: Life Science Alliance manuscript #LSA-2022-01771-T

Prof. Kazutoshi Mori
Kyoto University
Biophysics, Graduate School of Science
Kitashirakawa-oiwake
Sakyo-ku
Kyoto, Kyoto 606-8502
Japan

Dear Dr. Mori,

Thank you for submitting your manuscript entitled "Activation of XBP1 but not ATF6a Rescues Heart Failure Induced by ER Stress Lasting from Fertilization in Medaka" to Life Science Alliance. The manuscript was assessed by expert reviewer, whose comments are appended to this letter. We invite you to submit a revised manuscript addressing the Reviewer comments.

When submitting the revision, please include a letter addressing the reviewer's comments point by point.

Thank you for this interesting contribution to Life Science Alliance. We are looking forward to receiving your revised manuscript.

Sincerely,

B. MANUSCRIPT ORGANIZATION AND FORMATTING:

Reviewer #1 (Comments to the Authors (Required)):

In this manuscript, Jin and colleagues prepared and characterized a heart failure model in medaka fish through genetic deletion of the ER ATP/ADP exchange factor AXER. They then overexpressed either of the active UPR-associated transcription factors XBP1s or ATF6 to show that only overexpression of XBP1s rescued this heart failure defect allowing the fish to survive 3 days longer than controls. Using RNAseq then then profiled changes in gene expression in the heart to determine the basis of this protection. Then they compared gene expression changes observed in AXER-deficient fish, but rescued by XBP1s overexpression (not ATF6) to identify potential mechanisms that contribute to the observed heart failure model. They ultimately conclude that ER stress-dependent activation of multiple genes downstream of PERK-ATF4 are main factors driving pathology in this model, while XBP1s overexpression ameliorates ER stress and subsequent PERK activity.

From my perspective, the model is well-established/characterized on an organismal level and the RNAseq analysis appears well performed, providing a lot of information related to this model. However, there is a dearth of mechanism describing the relationship between ER stress, XBP1s overexpression, and heart failure in this model. For example, the conclusion is that PERK-ATF4 activity is a driver of pathology, but this is never tested. At the very least, this should be experimentally determined. Similarly, there is no effort to test the importance of specific XBP1s target genes in this pathology (although some potential candidates are identified). Apart from the lack of mechanism, the importance of the results are also unclear. AXER deficiency reduces ATP in the ER could directly influence the activity of key ER ATP-dependent chaperones and/or potential other mechanisms (e.g., Ca²⁺ regulation) to promote ER stress and heart failure. Again, these different mechanisms are not tested or worked out in this manuscript. This is important to better understand the underlying heart failure observed.

Ultimately, the push-pull for this manuscript comes from the dynamic between the experiments being well performed and the model being well-established/characterized vs. the lack of mechanistic insights into the underlying molecular events leading to pathology/rescue with XBP1s overexpression and the limited significance of the findings as described. Life Science Alliance does publish descriptive data of high quality that is important to the field, which could be an accurate characterization of this manuscript with some additional textual changes. Along those lines, this manuscript should be reworked to better describe the results shown and provide more caveats to address the mechanism of model/pathology/rescue and the significance of the findings in a larger context. That would increase the value to the community to meet the bar required to publish descriptive data in Life Science Alliance.

Kazutoshi Mori Ph.D.
Professor
Department of Biophysics, Graduate School of Science,
Kyoto University
Kitashirakawa-Oiwake, Sakyo-ku, Kyoto 606-8502, Japan
Tel: 81-75-753-4067, Fax: 81-75-753-3718
E-mail: mori@upr.biophys.kyoto-u.ac.jp

April 22, 2023

Dear Dr. Sawey,

Please find enclosed our manuscript entitled “Activation of XBP1 but not ATF6 α Rescues Heart Failure Induced by Persistent ER Stress in Medaka Fish” (Life Science Alliance manuscript #LSA-2022-01771-T), which has been revised in response to the reviewers’ comments, as detailed below.

We very much appreciate the valuable comments of the reviewers, which have helped us improve the quality of the manuscript considerably. We hope you will find the revised manuscript suitable for publication in LSA.

I look forward to hearing from you soon.

Sincerely,

Kazutoshi Mori Ph. D.

From my perspective, the model is well-established/characterized on an organismal level and the RNAseq analysis appears well performed, providing a lot of information related to this model. However, there is a dearth of mechanism describing the relationship between ER stress, XBP1s overexpression, and heart failure in this model. For example, the conclusion is that PERK-ATF4 activity is a driver of pathology, but this is never tested. At the very least, this should be experimentally determined.

We have removed the section “Induction pathway” from the revised manuscript, which will be reported somewhere after conducting additional experiments. Instead, we have shown that overexpression of 9 candidate gene products possibly associated with heart failure (marked by red circles in new Fig. 9) by mRNA microinjection into one-cell stage embryos of WT fish caused abnormal phenotypes of hearts (new Fig. 10A).

Similarly, there is no effort to test the importance of specific XBP1s target genes in this pathology (although some potential candidates are identified). Apart from the lack of mechanism, the importance of the results are also unclear. AXER deficiency reduces ATP in the ER could directly influence the activity of key ER ATP-dependent chaperones and/or potential other mechanisms (e.g., Ca²⁺ regulation) to promote ER stress and heart failure. Again, these different mechanisms are not tested or worked out in this manuscript. This is important to better understand the underlying heart failure observed.

We have shown that although transcription of ER chaperone genes was induced by ATF6 α (N), transcription of genes encoding ERAD components was induced by XBP1(S) but not by ATF6 α (N) using quantitative RT-PCR (new Fig. 10B) and discussed that “Critically, luminal ATP is consumed by ER chaperones but not by ERAD, because ATP-dependent ubiquitination and degradation by the proteasome occur in the cytoplasm (Depaoli et al., 2019). Thus, ER stress evoked by the decrease in luminal ATP level is ameliorated by ERAD of misfolded proteins accumulated in the ER, the capacity of which is enhanced by transcriptional induction of genes encoding ERAD components by the IRE1-XBP1 pathway” (p. 24, lines 9-14).

Ultimately, the push-pull for this manuscript comes from the dynamic between the experiments being well performed and the model being well-established/characterized vs. the lack of mechanistic insights into the underlying molecular events leading to pathology/rescue with XBP1s overexpression and the limited significance of the findings as described. Life Science Alliance does publish descriptive data of high quality that is important to the field, which could be an accurate characterization of this manuscript with some additional textual changes. Along those lines, this manuscript should be reworked to better describe the results shown and provide more caveats to address the mechanism of model/pathology/rescue and

the significance of the findings in a larger context. That would increase the value to the community to meet the bar required to publish descriptive data in Life Science Alliance.

We believe that the above mentioned two additional experiments have provided mechanistic insights into the underlying molecular events leading to pathology/rescue with XBP1s overexpression (we would say full activation of XBP1 rather than overexpression), which have sufficiently enhanced the significance of our findings.

April 24, 2023

RE: Life Science Alliance Manuscript #LSA-2022-01771-TR

Prof. Kazutoshi Mori
Kyoto University
Biophysics, Graduate School of Science
Kitashirakawa-oiwake
Sakyo-ku
Kyoto, Kyoto 606-8502
Japan

Dear Dr. Mori,

Thank you for submitting your revised manuscript entitled "Activation of XBP1 but not ATF6a Rescues Heart Failure Induced by Persistent ER Stress in MedakaFish". We would be happy to publish your paper in Life Science Alliance pending final revisions necessary to meet our formatting guidelines.

- please relabel your supplementary figures as Figure S1, Figure S2, etc. in both your figure legend and in the figure callouts in the main manuscript text
- please add the Twitter handle of your host institute/organization as well as your own or/and one of the authors in our system
- please add the author contributions and a conflict of interest statement to the main manuscript text
- please use the [10 author names, et al.] format in your references (i.e. limit the author names to the first 10)
- please add a Data Availability statement indicating accession information for readers to access the RNA-seq dataset

Figure Check:

- please add scale bars to Figure 3C
- the file "Figure 3-video" should be renamed as a supplemental figure if you want to keep this as showing the stills from the videos, the Videos will need their own legends

A. FINAL FILES:

B. MANUSCRIPT ORGANIZATION AND FORMATTING:

Thank you for your attention to these final processing requirements. Please revise and format the manuscript and upload materials within 3 days.

Sincerely,

April 26, 2023

RE: Life Science Alliance Manuscript #LSA-2022-01771-TRR

Prof. Kazutoshi Mori
Kyoto University
Biophysics, Graduate School of Science
Kitashirakawa-oiwake
Sakyo-ku
Kyoto, Kyoto 606-8502
Japan

Dear Dr. Mori,

Thank you for submitting your Research Article entitled "Activation of XBP1 but not ATF6a Rescues Heart Failure Induced by Persistent ER Stress in MedakaFish". It is a pleasure to let you know that your manuscript is now accepted for publication in Life Science Alliance. Congratulations on this interesting work.

DISTRIBUTION OF MATERIALS:

Again, congratulations on a very nice paper. I hope you found the review process to be constructive and are pleased with how the manuscript was handled editorially. We look forward to future exciting submissions from your lab.

Sincerely,
